# Genetic Variability of *Tabebuia rosea* (Bignoniaceae) from Plantations and Remnant Populations in the Mayan Forest

**Hugo Ruiz-González** [1,†]**, María Raggio** [1,†]**, Natalia Y. Labrín-Sotomayor** [1]**, Miriam M. Ferrer** [2,*]
**and Yuri J. Peña-Ramírez** [1,*]

1   Departamento de Ciencias de la Sustentabilidad, El Colegio de la Frontera Sur unidad Campeche,
    Campeche 24500, Mexico; hugo.ruiz@estudianteposgrado.ecosur.mx (H.R.-G.);
    maraggio@ecosur.edu.mx (M.R.); nlabrin@ecosur.mx (N.Y.L.-S.)
2   Departamento de Manejo y Conservación de Recursos Naturales Tropicales, Campus de Ciencias Biológicas y
    Agropecuarias, Universidad Autónoma de Yucatán, Mérida 97313, Mexico
*   Correspondence: mferrer@correo.uady.mx (M.M.F.); ypena@ecosur.mx (Y.J.P.-R.);
    Tel./Fax: +52-999-9423212 (M.M.F.); +52-981-1273720 (Y.J.P.-R.)
†   These authors contributed equally to this work.

**Abstract:** In Neotropical regions, plantations and remnant forest populations of native trees coexist in a highly fragmented matrix and may be affected by isolation and reduction in population size, leading to genetic structure, inbreeding, and genetic bottlenecks that reduce the population's genetic diversity. *Tabebuia rosea* variability in the Mayan Forest was studied by genotyping 30 trees from three plantations and three remnant natural populations using simple sequence repeats (SSRs) and inter-simple sequence repeats (ISSRs). $H_o$-SSR estimates were lower than $H_e$; the mean inbreeding coefficient was 0.07 and did not differ among populations, but was eight times higher in plantations than in remnant populations. Using ISSR data, the individuals were assigned to k = 5 and k = 4 clusters under admixture without and with geographic information used as priors in Bayesian analysis assignments. Genetic differentiation estimated with the Bayesian estimator II (0.0275 ± 0.0052) was significantly different from 0, but $F_{ST}$ was not (0.0985 ± 0.1826), while paired $F_{ST}$ among populations ranged from 0.05 up to 0.16. Only one remnant population displayed evidence of a genetic bottleneck. *T. rosea* displays a genetic structure in which the isolated remnant forest populations show moderate inbreeding levels.

**Keywords:** genetic diversity; genetic structure; forest genetic resources; tropical timber species





## 1. Introduction

Genetic variability occurs at different spatial and temporal scales due to the interactions among microevolutionary forces and ecological and anthropic factors, affecting the distribution patterns of gene and allele frequencies [1]. In the Neotropical region, particularly in Mesoamerica, most remnant forest fragments are immersed in a matrix where traditional agroforestry and agricultural systems coexist with monoculture plantations and urban vegetation [2]. In those systems, native forest germplasm has been managed since prehispanic times [3]. The Mayan Forest is highly fragmented due to intense deforestation cycles driving forest populations of native plants to local extinction or confining them to secondary vegetation [4,5]. Therefore, native wild populations may have experienced genetic bottlenecks caused by extensive logging, thinning, or natural disasters that may reduce genetic diversity [6,7]. When isolation increases, as a consequence of fragmentation, inbreeding may hasten, too [8,9].

In the case of commercial plantations, the provenance and number of selected mother plants [10,11], the seed collection method (random or systematic) [10,12], and the nursery practices [13,14] define the genetic diversity of the stands. A reduction in genetic diversity compared to native populations is expected if seed collection results from relatively few

sources [13,15] or when donor trees are chosen by productivity traits resulting in directional selection [16]. Constant gene flow driven by high outcrossing rates is a typical characteristic of tropical trees, allowing the maintenance of high levels of genetic diversity and preventing drifts in spatial genetic structure [17,18]. Population differentiation is expected when the isolation of populations is enough to reduce gene flow via pollen or seed dispersion [19,20].

*Tabebuia rosea* (Bertol.) Bertero ex A.DC. is a native tree from tropical America distributed from Brazil to Mexico, including Central America and the Antilles [21]. *T. rosea* is found in commercial plantations mainly for high-quality timber production and urban and peri-urban plantations due to its colorful blossoms and ecosystemic services [22]. In Mexico, more than half of the plantations of this species are established in the southern states, including the Yucatan Peninsula [23]. Moreover, in this region, the species is widely distributed; it can be found in primary or secondary vegetation patches, and also in agroforestry systems such as dispersed trees in pasturelands and home gardens [24,25]; these types of arrangements usually make the detection of genetic variability reduction difficult because of the presence of the previously fragmented population [26]. Therefore, as commonly occurs in other forestry species [11,27], *T. rosea* is found in plantations alongside the natural distribution range of native populations, and the gene pool may consist of both reservoirs that should be maintained by their interactions [28].

This study aimed to evaluate the genetic variability in six populations from plantations and remnant native populations of *T. rosea* using SSR [29,30] and ISSR [31] molecular markers. Both plantations and remnant fragments coexist along the Calakmul–Bala'an K'aax biodiversity corridor, formerly part of the Mesoamerican Biological Corridor [32], in a highly fragmented landscape. We decided to use ISSR and SSR molecular markers as both have demonstrated efficiency in population genetic analysis [33]. ISSR markers require small DNA quantities; sequence data for primer construction is unnecessary, and amplicon loci are randomly distributed throughout the genome, generating many polymorphic bands per reaction, and reliable and reproducible results [34]. On the other hand, SSR markers have previously been developed for the genus and the species [29,31], easing their applicability due to their co-dominant nature and high polymorphism; comparing different species populations is insightful.

## 2. Materials and Methods

### 2.1. Study Site

This study was conducted in the western area of the Yucatan Peninsula, Mexico, encompassing most of the Karst and Hills of Campeche and surrounding physiographic provinces [35]. An Aw2 climate characterizes this region with an average of 1000 to 1400 mm annual rain and 26.2 °C mean temperature, with two months of the dry season followed by a summer rainy season [36]. Three populations, delivered sown for either reforestation or timber extraction purposes, were included and are referred to as plantations hereafter. The three remnant populations were found either in pasturelands for livestock production or in secondary vegetation (remnants hereafter). The plantations comprised from 0.5 to 120 ha, harboring high population density compared to remnant populations, which consisted of 34.92 to 269.79 ha (Table 1). The "Ck" plantation is a six-year-old suburban multispecies reforestation area including *Cedrela odorata* L., *Swietenia macrophylla* King., *Azadirachta indica* A. Juss., *Delonix regia* Hook. Raf., *Ceiba pentandra* L., and *T. rosea*, located in the peri-urban area of *Calkiní* town (Figure 1). The "Cp" plantation is a reforested area within *San Francisco de Campeche* city, in which monospecific *T. rosea* plantation is maintained as ornamental (Figure 1). The "Xb" plantation is a nine-year-old commercial timber plantation for producing *T. rosea*, *C. odorata*, and *S. macrophylla* near the *Xbacab* community (Figure 1). The "Dn" remnant population was found in a secondary vegetation area surrounding rural settlements located in *División del Norte* community (Figure 1). The "At" remnant population was found in a remnant forest fragment of medium-height sub-deciduous near *Atasta* town (Figure 1). The "Co" remnant population was found in pasture land where individuals of *T. rosea* are tolerated sensu near *Constitución* community.

**Table 1.** Population density, patch size, and sampled area of the *Tabebuia rosea* populations were sampled in a fragmented landscape of the Mayan Forest in the Yucatan Peninsula.

| Location | Type | Patch Area (ha) | Sampled Area (m²) | Estimated Individuals per ha | Average Distance between Sampled Individuals (m) |
|---|---|---|---|---|---|
| ● Ck | Plantation | 6.0 | 13,780 | 23.22 | 23.76 ± 15.46 |
| ● Cp | Plantation | 0.5 | 49,200 | 65.04 | 17.59 ± 15.29 |
| ● Xb | Plantation | 120.0 | 120,000 | 71.11 | 69.91 ± 51.41 |
| ■ Dn | Remnant | 269.79 | 332,760 | 0.96 | 286.58 ± 409.87 |
| ■ At | Remnant | 66.80 | 104,760 | 3.05 | 92.44 ± 110.48 |
| ■ Co | Remnant | 34.92 | 113,080 | 2.83 | 94.24 ± 133.45 |

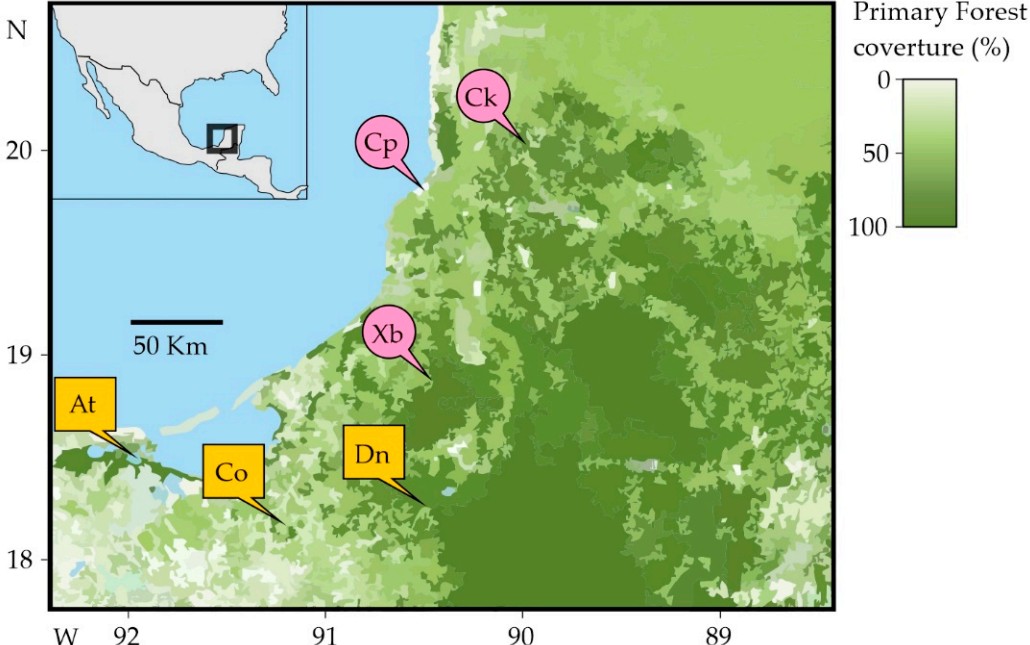

**Figure 1.** Location of sampled populations in a fragmented landscape of the Mayan Forest. The three plantations (round pink labels) are Ck, Cp, and Xb, and the three remnant populations (square yellow labels) are At, Co, and Dn, located in Campeche state in the Yucatan Peninsula.

### 2.2. Sampling Strategy

Adult, representative individuals having diameters at breast height higher than 35 cm were included in the sampling. Due to differences in the population density between plantation and remnant populations (Table 1), sampling strategies differ. Random sampling was used in plantations, whereas systematic selection was in the remnant populations. In the plantations, 30 random individuals were selected from transects along the entire Ck and Cp populations and 1/3 of the Xb population. In the remnant populations, walks through the inner paths were performed until 30 adults were found. Five to ten young leaves were collected from the middle section of the crown and placed in a plastic zipper bag with 100 g of silica gel, transported in a cooler at 4 °C to the laboratory, and stored at −70 °C until processing.

### 2.3. DNA Isolation, SSR, and ISSR Scoring Analyses

To isolate genomic DNA, 1.5 g of leaf tissue was ground in liquid nitrogen and stored at −20 °C. According to the manufacturer's protocol, the grounded tissue was weighed, and 50 mg were processed for DNA extraction with the Quick-DNA Plant/Seed Miniprep Kit (Cat. # 6020, Qiagen®, Hilden, Germany). The quality of genomic DNA was confirmed visually by electrophoresis in 0.8% agarose gels. The 18 microsatellites (SSR) primer

combinations [29,30] were evaluated using the original protocols to amplify DNA from 10 random samples. Only TRB6, TRA101, TRC103, TRB104, TRC105, TRA109, TRB109, Tau22, and Tau31 [31] yielded enough polymorphic bands to be considered for this work. A modified PCR reaction and program were used to obtain the SSR amplicons as follows: 15 μL reaction volume containing 1X Taq Buffer with KCl, 1.5 mM $MgCl_2$, 0.375 U Taq DNA polymerase recombinant (Thermo Scientific, Carlsbad, CA, USA), 0.2 mM dNTPs, 0.2 μM each primer, 20 ng of template, 1% BSA, and $ddH_2O$. Amplification was conducted on a Select Cycler II thermal cycler (Cat. # SBT9610. Select BioProducts Edison, NJ, USA) with the following conditions: initial denaturing step at 95 °C for 10 min, followed by 30 cycles of 94 °C for 1 min, 56 °C for 1 min, and 72 °C for 1 min, and then a final extension step of 72 °C for 5 min. For ISSR analysis, four primers from the British Columbia University database were employed based on the number of polymorphic bands obtained (UBC-835, UBC848, UBC856, and UBC 891) from an evaluation of the 12 primers from UBC set 9. PCR products from ISSR were obtained using a 15 μL reaction volume containing 1X Taq Buffer with KCl, 1.5 mM $MgCl_2$, 0.375 U Taq DNA polymerase recombinant (Thermo Scientific, Carlsbad, CA, USA), 0.2 mM dNTPs, 1.0 μM ISSR primer, 20 ng of template, 1% BSA, and $ddH_2O$. Amplification was conducted under the following conditions: initial denaturing step at 95 °C for 10 min, followed by 30 cycles of 94 °C for 1 min, 58 °C for 1 min (UBC835 and UBC891), or 60 °C for 1 min (UBC848 and UBC856), and 72 °C for 2 min, followed by an extension step of 72 °C for 10 min. Amplified bands were confirmed by 1.5% agarose electrophoresis. Amplicon size from SSR and ISSR was determined by capillary electrophoresis using the QIAxcel DNA High-Resolution Gel Cartridge and the QX size marker 25–500 bp in a QIAxcel instrument (Cat # 9001941, Qiagen IPA, Hilden, Germany).

### 2.4. Data Analyses

Two data matrixes were constructed for each molecular marker (Supplementary File S1), including 180 individuals (30 per population). Amplicon size was recorded per individual to account for different alleles on the nine SSRs loci markers. The presence and absence of all clear bands of different sizes were recorded for an individual to account for other loci for each of the four ISSR markers amplified. Null allele presence per population was determined using Micro Checker (University of Hull, Kingston, UK) [37]. Due to more than 10% of the genotypes bearing potentially null alleles for most of the SSRs markers, allele frequency was not calculated with this program. To evaluate the occurrence of inbreeding, failure of genotypes, and null alleles, INEST v. 2.2 (Kazimierz Wielki University, Bydgoszcz, Poland) [38] was used. The complete model and the one with failure in genotype and inbreeding had lower log-likelihood values for the six populations and total data. Both models did not differ in more than two units of the log-likelihood values at the population and total data. The genetic structure posterior probabilities and the genetic differentiation were not analyzed with the SSRs markers because the high prevalence of null alleles prevents the correct identification of genotypes required to obtain the similarity–differentiation statistics.

### 2.5. Genetic Structure

The genetic structure of the populations was assessed with ISSR markers separately using a Bayesian assignment of individuals to k genetic clusters [39] and the analysis of population structure with the coordinates per individual as prior information [40]. The k genetic clusters were determined by the comparison of the log-likelihoods estimated from ten iterations per k value from 1 to 20 clusters under a mixture and admixture model with Structure v. 2.3.4 (University of Oxford, Oxford, UK) [39] (Supplementary File S2) and for the spatial mixture and admixture model with BAPS v. 6.0 (University of Lausanne, Lausanne, Switzerland) [40]. The log-likelihoods of belonging to k clusters were evaluated with Harvest structure for all 200 iterations obtained with Structure v. 2.3.4 [39] and those from the best ten partitions according to BAPS [40]. Then, values from the partitions obtained with the structure for the most likely value of *k* were clumped using the Greedy

algorithm and 100,000 permutations [41]. The partition from the admixture model with the posterior probabilities of belonging to a k cluster was annotated by combining the individuals from the six populations and whether they were from plantation or remnant populations using Distruct v. 1.1 (University of Connecticut, Storrs, CT, USA) [42].

### 2.6. Genetic Differentiation and Genetic Diversity

Genetic differentiation among the six populations was evaluated with the Bayesian estimator II [43], and the global and paired fixation index ($F_{ST}$) values, and at the individual level with the differentiation index (1-relatedness) among all individuals of the six populations [44] with the ISSR markers. For the Bayesian approach, four models were compared using the deviance information criterion (DIC) obtained with Hickory v.1.1 (Free Software Foundation: Boston, MA, USA) [45]. These models differ by the inclusion of both inbreeding coefficient estimate ($f$) and $\Theta_{II}$ (full model), exclusion of $f$ ($f = 0$ model), exclusion of $\Theta_{II}$ ($\Theta = 0$), and free f estimation per population (f free model), and from those, the model had the lowest value for the DIC with a difference of more than ten units from the second lower value. The mean, standard error, and lower and upper limits of $F_{ST}$ were estimated jackknifing over loci in 1000 permutations, while 1000 bootstrapped paired $F_{ST}$ values were obtained with AFLP-Surv v.1.0 (Université Libre de Bruxelles, Brussels, Belgium) [46] using the $F_{is}$ estimator for the total data. To heuristically test (a) the genetic differentiation among the six populations, paired with the $F_{ST}$ values, and (b) the dissimilitude among all individuals, the six-two UPGMA dendrograms were constructed using PHYLIP v.3.2 [47] and edited with MEGAX v.11 (The Pennsylvania State University, PA, USA) [48]. The genetic diversity per population, plantation, remnant populations, and each cluster identified in the genetic structure analyses were estimated with the number of alleles and observed and expected heterozygosity ($H_o$ and $H_e$, respectively) for the SSR markers with the aid of INEST v.2.2 [38]. The percentage of polymorphic loci (PPL) and $H_e$ [44] for ISSR markers was tested with the aid of AFLP-Surv v.1.0 [46]. Due to the dominant nature of ISSRs, allele frequencies were estimated using a Bayesian approach [49]. The average $H_o$ and $H_e$ per population and cluster were compared using a Tukey–Kramer HSD post hoc test.

### 2.7. Genetic Bottlenecks

To test for recent bottlenecks in each population, the estimates of the expected heterozygosity under mutation–drift equilibrium ($H_{eq}$) were compared with the $H_e$ [50]. INEST v.2.2 was used to perform this comparison under the two-phase mutation model for the SSR markers., and its significance was evaluated using the Z test on combined Z scores per loci and Wilcoxon signed-rank test from which *p* values were determined both assuming normal distribution and from 1000 permutations to approximate the exact value.

### 3. Results

The 180 sampled individuals were assigned to k = 3 clusters, assuming a mixture model for the six populations, to k = 5 clusters, assuming admixture (both with structure v.2.3.4 [39]), and to k = 4 clusters using the geographical location of each individual, assuming mixture and admixture (with BAPS v.6.0 [51]). Only the model without geographical information a priori suggests the presence of admixture in Dn and At remnant populations and, to a lesser degree, among individuals from Ck and Cp plantations (Figure 2). The assignation of individuals in the admixture model with geographical information a priori included for cluster 1 (purple in Figure 2): all the individuals from Xb plantation and Co remnant population; cluster 2 (red in Figure 2): 29 of the individuals from Dn and all from At remnant populations; cluster 3 (orange in Figure 2): all from Ck plantation; cluster 4 (green in Figure 2): 29 from Cp plantation; and cluster 5 (cyan in Figure 2): in admixture individuals from Ck, Cp, At, and Dn. According to the admixture analysis of BAPS, one individual from the Cp plantation and one from Dn remnant populations had

mixed assignations. Estimated $\Theta_{II}$ (0.0275 ± 0.0052) differed significantly from 0, but FST (0.0985 ± 0.1826) did not.

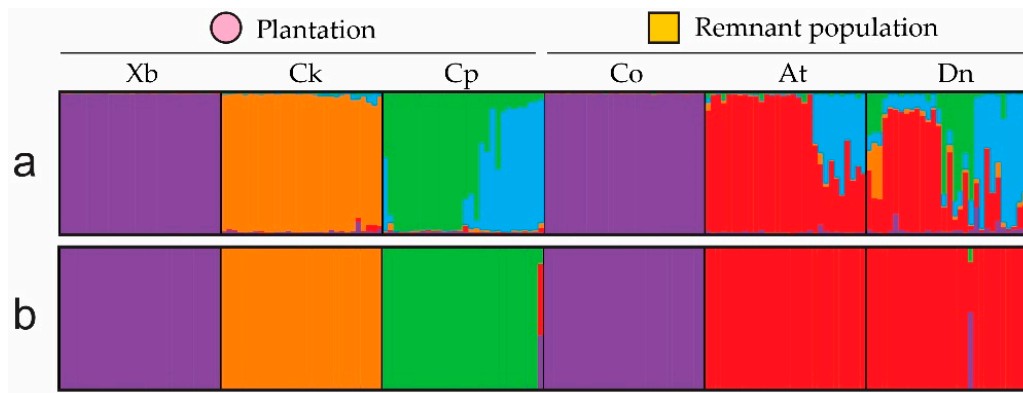

**Figure 2.** Posterior probabilities of belonging to k clusters assuming admixture from the Bayesian assignment analyses (**a**) without a geographic location a priori using structure v.2.3.2, and (**b**) with geographic location a priori using BAPS v. 6. The colors used were purple for cluster 1, red for cluster 2, orange for cluster 3, green for cluster 4, and cyan for cluster 5, grouping individuals mostly from two plantations and two remnant populations.

The paired FST among populations ranged from 0.05 to 0.16; the more significant differences were found among the Co remnant population and the Xb plantation, followed by the remaining four populations (Figure 3). Cp and Ck plantations were grouped with the remnant populations from Dn and At with high bootstrap support, although the inner grouping had low bootstrap support (Figure 3). Like the Bayesian assignments and the paired FST dendrogram, the genetic differences among individuals suggest a robust genetic structuring in which Co and Xb are a genetic group closely related, and all individuals from the remaining four populations are closely related to others from the same locality they inhabit (Figure 3b).

Using SSR markers, most individuals had an identifiable genotype, and the number of alleles ranged from 3 in the Xb plantation and the Co remnant population up to 16 in Cp plantation and Dn remnant populations (Table 2). All the populations and clusters had a lower $H_o$ than the $H_e$ (Table 2). Therefore, the average inbreeding coefficient was 0.07 and varied from 0.03 to 0.08 in the At remnant population and Ck plantation, respectively, and from 0.06 to 0.28 in clusters 4 and 2, respectively. In plantations, the inbreeding coefficient was almost eight times higher than in remnant populations. The 95% confidence intervals of the inbreeding coefficients among populations and clusters overlapped but not for the comparison among plantations and remnant populations (Table 2).

Using ISSR markers, the average PPL was 32.06 (ranging from 8.8 to 49.3), and the $H_e$ was 0.0866 ± 0.0069 (ranging from 0.0243 ± 0.0046 to 0.1309 ± 0.0074). The NPL and the PPL were lower, and the $H_e$ was slightly higher in plantations compared to remnant populations (Table 2). The PPL was two times lower in the Xb plantation and the Co remnant population compared to the other four populations; similarly, differences in the average $H_e$ for those populations had three to five times lower values compared with the remaining four populations. Consequently, cluster 1, in which most of the individuals of those populations were assigned, also had three to four times lower values compared to the other clusters for either index (Table 2). Only in the Dn remnant population, which had the lowest density among all six populations (Table 1), was a bottleneck detected with comparisons for the $H_{eq}$ and the $H_e$.

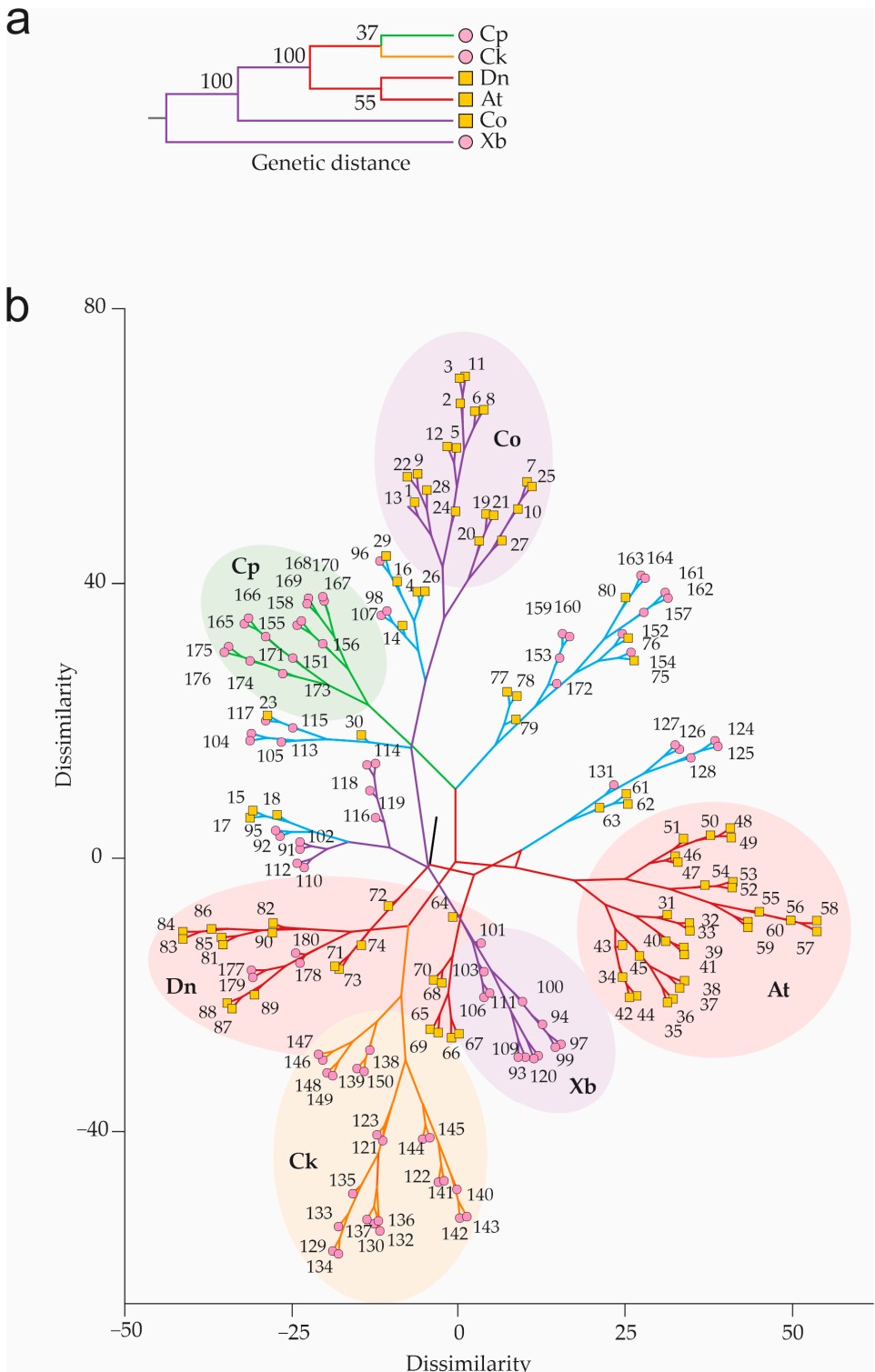

**Figure 3.** Genetic differentiation is depicted in (**a**) a dendrogram for 1000 bootstrap values of the paired FST among six *T. rosea* by UPGMA clustering at the population level and (**b**) at the individual level (complete method). Symbols and branch colors depict the five genetic clusters according to the Bayesian assignment of population structure from Figure 2: cluster 1 (purple), cluster 2 (red), cluster 3 (orange), cluster 4 (green), and cluster 5 (cyan). Round pink circles correspond to plantation populations, whereas yellow squares belong to remnant populations. Numbers in the tree correspond to 1–30 for Co, 31–60 for At, 61–90 for Dn, 91–120 for Xb, 121–150 for Ck, and 151–180 for Cp populations. Colored shadows in clusters for populations correspond to clusters 1–4.

**Table 2.** Genetic diversity indexes were calculated from nine SSRs loci (NG, number of genotypes; NA, number of alleles; Ho, observed heterozygosity; He, expected heterozygosity; and F, inbreeding coefficient (third to seventh column)) and from 296 ISSRs loci (NPL, number of polymorphic loci; PPL, percentage of polymorphic loci; and He, expected heterozygosity (eighth to tenth column)). Data were gathered from 180 individuals of *Tabebuia rosea* from the Mayan Forest from the Yucatan Peninsula. Individuals (a) were grouped in populations, (b) population types (plantations and remnant populations), and (c) clusters generated by the BAPS model.

| Grouping | n | NG | NA | $H_o$ (SE.) | $H_e$ (SE.) | F (95% CI) | NPL | PPL | $H_e$ (SE.) |
|---|---|---|---|---|---|---|---|---|---|
| **(a) Population** | | | | | | | | | |
| • Ck | 30 | 26 | 15 | 0.760 [b] (0.133) | 0.840 [b] (0.133) | 0.08 (0.002–0.20) | 114 | 38.5 | 0.105 [b] (0.008) |
| • Cp | 30 | 23 | 16 | 0.774 [b] (0.156) | 0.838 [b] (0.150) | 0.06 (0.001–0.18) | 146 | 49.3 | 0.131 [b] (0.007) |
| • Xb | 30 | 23 | 3 | 0.416 [a] (0.234) | 0.521 [a] (0.212) | 0.07 (0.002–0.18) | 26 | 8.8 | 0.024 [a] (0.005) |
| ■ Dn | 30 | 25 | 16 | 0.804 [b] (0.169) | 0.854 [b] (0.168) | 0.06 (0.001–0.16) | 126 | 42.6 | 0.106 [b] (0.007) |
| ■ At | 30 | 26 | 12 | 0.752 [b] (0.166) | 0.802 [b] (0.121) | 0.03 (0.000–0.10) | 129 | 43.6 | 0.118 [b] (0.008) |
| ■ Co | 30 | 24 | 3 | 0.418 [a] (0.232) | 0.474 [a] (0.251) | 0.06 (0.002–0.16) | 113 | 38.2 | 0.036 [a] (0.006) |
| **(b) Population type** | | | | | | | | | |
| • Plantation | 90 | 89 | 23 | 0.451 (0.149) | 0.880 (0.084) | 0.47 (0.435–0.51) | 38 | 12.8 | 0.093 (0.005) |
| ■ Remnant | 90 | 76 | 22 | 0.787 (0.093) | 0.860 (0.100) | 0.06 (0.003–0.18) | 117 | 39.5 | 0.092 (0.006) |
| **(c) Clusters** | | | | | | | | | |
| Cluster 1 | 60 | 59 | 4 | 0.374 [c] (0.141) | 0.552 [c] (0.196) | 0.27 (0.055–0.39) | 39 | 13.2 | 0.031 [c] (0.005) |
| Cluster 2 | 60 | 60 | 20 | 0.594 [d] (0.206) | 0.849 [d] (0.142) | 0.28 (0.192–0.34) | 131 | 44.3 | 0.115 [d] (0.007) |
| Cluster 3 | 30 | 26 | 15 | 0.764 [d] (0.133) | 0.840 [d] (0.133) | 0.08 (0.002–0.20) | 114 | 38.5 | 0.105 [d] (0.008) |
| Cluster 4 | 30 | 23 | 16 | 0.774 [d] (0.156) | 0.838 [d] (0.150) | 0.06 (0.001–0.18) | 146 | 49.3 | 0.131 [d] (0.007) |

Different letters among the $H_o$ and $H_e$ values were significantly different according to the Tukey–Kramer HSD post hoc test among the six populations and four clusters. Admixture cluster 5 was not considered.

## 4. Discussion

The plantations and remnant populations of *T. rosea* that currently coexist in the Mayan Forest display low to moderate levels of inbreeding, genetic diversity, and genetic differentiation because of their management. Biparental inbreeding and founder effects are related to the genetic structure of the plantations and remnant populations in this fragmented landscape. Germplasm wild lineages are maintained in plantations, although they are less diverse and more inbreeding than those from wild remnant populations.

The analyzed *T. rosea* populations presented moderate but significant levels of inbreeding; most inbreeding coefficients estimated correspond to those from mating among first cousins, suggesting that recruitment of related individuals is favored within populations. In outcrossing woody plants biotically pollinated, inbreeding is expected to be low [9]. However, species harboring low-density populations in tropical forests, mass-blooming seedlings, and isolation may hasten local mating in the fragmented populations [52]. High inbreeding coefficients appear as a constant in the Tabebuia genus and a wide range of population types, including a highly impacted population [53] and a continuous population [28]. In the cases mentioned above, high levels of inbreeding were associated with the species' short flower display period; this is also true for *T. rosea*, which blooms for less than two months during the dry season in the study area. When combined with the isolated patchy distribution of the species, this flowering behavior may favor biparental inbreeding [54,55] because bee foragers tend to visit flowers from closely established individuals [8,54]. Breeding among close relatives can decrease the genetic diversity within populations and favor the genetic structure among plantations and remnant *T. rosea* populations. It is not uncommon that some tropical woody plants display biparental behavior [55].

The genetic structure of the species observed in this study, composed of three to five genetic clusters, displays some admixture levels, except for Xb and Co, which pertain to just one cluster. As expected, the clustering models showed distinct genetic structures depending on using geographic information per individual a priori in the Bayesian assignation. Geographic information is relevant for the remnant populations in which local

recruitment is expected; however, due to the management practices in urban and commercial plantations, recruitment is not allowed, and young trees from the same coming from different nurseries may be sown, allowing admixture, as shown in the analyses in which non-priori geographic coordinates were used. The following paragraphs will discuss the pattern arising from the latter analyses.

The clear distinction of each plantation and remnant populations portraying one or few lineages may be caused by biparental inbreeding, as discussed before, and by a founder effect in plantations. Plantations and remnant populations belonging to the same clusters may be an effect of the propagation of the species in local nurseries, which gather seeds from a few mother trees that may become the most abundant germplasm lines of the species, even if the genotype was underrepresented in remnant populations, as was observed for cluster 3. This result is similar to that found in Araucaria angustifolia Bertol. Kuntze plantations that were established from plants bought at local nurseries (which usually obtain their seeds from wild populations) [56,57]. The assignation in clusters and the grouping in the UPGMA dendrograms suggest that individuals from the tree plantations were propagated from regional germplasm, although each came from different lineages.

The genotype admixture of some of the clusters from one plantation and two remnant populations suggests that those populations can arise from a larger population in which more genetic variability was granted by higher gene flow. The clusters and groups in which both plantations and remnant populations are admixture highlight the importance of the remnant forest fragments and pasturelands as germplasm sources that are locally used in reforestation programs or as germplasm sources in sylviculture activities. The *T. rosea* Cp plantation is a discrete reservoir of at least two genetic wild lineages present in the physiographic region. In contrast, the other two are the reservoirs or one wild lineage each. Given that some individuals were admixture in plantations, genotypic combinations of the wild lineages are also present in plantations. Further efforts to evaluate the provenance of the individuals in plantations may strengthen the role that the use of native trees in urban reforestation programs plays in forest germplasm conservation.

Plantations and remnant forest populations in secondary vegetation and pasturelands are the main population types for *T. rosea* found in the biological corridors of Southern Tropical Mexico. The intentional planting of trees in commercial and urban plantations guarantees gene flow, primarily through seedlings grown in nurseries, as discussed before, even when population distances span more than 290 km (Supplementary File S1). Additionally, the presence of some scattered trees in home gardens, life fences, and parks in the fragmented landscape that characterize this region [25,26] may facilitate the connectivity of the more conspicuous plantations and remnant populations we studied here, as has been suggested for three other native tropical trees of Southern Mexico [57].

This study found that the plantations maintain similar allelic richness and heterozygosis levels to the closest remnant populations of *T. rosea*. The allelic and genetic diversity values were similar to those of other species of the genus in which SSRs molecular markers were used. For the Tabebuia genus, previous studies have demonstrated that the genetic diversity is similar for several populations: between urban plantations and secondary vegetation in *T. roseo-alba* (Ridl.) sand from Brazil's Cerrado [53], between continuous and fragmented landscapes in Brazil's Cerrado for *T. ochraceae* A. H. Gentry [28], and between seed orchards and progeny trials from different provenances in Colombia for *T. rosea* [30]. The estimates of allelic and genetic diversity for the plantation and remnant population from cluster 1 in this study (Table 2) were among the lowest values published for both the genus [29,31,58] and the species [30]. Estimates from $H_o$ and $H_e$ of cluster 1 were similar to those found in a fragmented and isolated population of *T. ochracea* in Brazil's Cerrado [28]. The PPL and $H_e$ values were like those estimated with ISSRs markers in remnant forest populations and Mayan home gardens of two dioecious species, *Brosimum alicastrum* Sw. and *Spondias purpurea* L., and a heterostylous one, *Cordia dodecandra* A. DC.; all of them species managed by Mayans [59]. Fragmentation and selective logging contribution to

decreasing genetic diversity should be tested in future works with this species, including a large number of populations.

## 5. Conclusions

Trees usually have large population sizes, high rates of gene flow, long lifespans, and preferential outcrossing mating systems [60]; as a result, genetic bottlenecks rarely occur, even in fragmented landscapes [61,62] or under intense logging regimes [63]. Even the most logged species in Brazil's seasonally dry forests, Handroanthus impetiginosus Mart. ex DC. Mattos, showed no genetic bottlenecks [58]. In this study, one remnant population (Dn) may have undergone a recent genetic bottleneck. This *T. rosea* population has 1/3 of the density population of the other remnant populations and two orders of magnitude lower density than the plantations. The reduction in size and density in remnant populations due to soil change, uncontrolled logging, and deforestation [3] raises concern about the conservation status of *T. rosea* in the Mayan Forest, in which populations sustain only low to moderate genetic diversity and biparental inbreeding. The plantations are only discrete reservoirs of the genotypic variability of the current populations. Due to new business opportunities for certified seed production and the urgency of the conservation of *T. rosea*, we recommend continuing studies on genetic variability in native populations of the region.

**Supplementary Materials:** The following supporting information can be downloaded at: https://www.mdpi.com/article/10.3390/f14102006/s1.

**Author Contributions:** Conceptualization, Y.J.P.-R. and M.M.F.; methodology, H.R.-G. and M.R.; software, H.R.-G., M.R., M.M.F. and Y.J.P.-R.; validation, N.Y.L.-S.; formal analysis, H.R.-G., M.M.F., N.Y.L.-S. and Y.J.P.-R.; investigation, H.R.-G. and M.R.; resources, M.M.F. and Y.J.P.-R.; data curation, H.R.-G. and M.R.; writing—original draft preparation, M.M.F. and Y.J.P.-R.; writing—review and editing, all authors; visualization, N.Y.L.-S., M.M.F. and Y.J.P.-R.; supervision, M.M.F. and Y.J.P.-R.; project administration, Y.J.P.-R.; funding acquisition, M.M.F. and Y.J.P.-R. All authors have read and agreed to the published version of the manuscript.

**Funding:** Funding from Research Grants: 5103711808 2013–2015 and 2019–2020 from El Colegio de la Frontera Sur to Y.J.P.-R., Ciencia Básica 169336-B and A1S-30471 from Consejo Nacional de Ciencia y Tecnología to M.M.F., and Master scholarship 812422 to H.R.-G., and 578607 to M.R.

**Data Availability Statement:** Crude data are available as Supplementary File S1.

**Acknowledgments:** Authors thank Gabriel Chan-Coba and Joaquin Gómez for field work assistance, and Leonel Villatoro de la Cruz from Agros Marañon SA de CV, Antonio Álvarez-Torres, and José Domingo Pérez-Vázquez from Rancho Don Bartolo for granted permission to work on their properties and plant material collection.

**Conflicts of Interest:** The authors declare no conflict of interest.

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
