# Peer review of "Genetic Variability of Tabebuia rosea (Bignoniaceae) from Plantations and Remnant Populations in the Mayan Forest"

_forests, doi:10.3390/f14102006_

Round 1

Reviewer 1 Report

Dear authors

First, I thank the authors for this interesting paper entitled” Genetic variability of Tabebuia rosea (Bignoniaceae) from plantations and remnant populations in the Mayan forest.

1- Introduction: it needs to add a paragraph about the importance of ISSR molecular markers in population genetics and add references and also the difference in the efficiency in your data between the ISSR and SSR results

Materials: 18 primers from SSR and 4 from ISSR are few, and you can add more primers to give more efficient diversity among the studied populations.

2- Results: we need to make a relationship between ISSR and SSR data to give a cluster tree for the 6 populations.

Can you send in the revised version Excel sheet for the data from ISSR and SSR used for structure software?

Some corrections in the revised paper in the attachment                                  

Author Response

Reviewer 1:

1- Request: Introduction: it needs to add a paragraph about the importance of ISSR molecular markers in population genetics and add references and also the difference in the efficiency in your data between the ISSR and SSR results

Answer: A section in the introduction, fourth paragraph has been added including requested information (Lines 66-72).

2-. Request: Materials: 18 primers from SSR and 4 from ISSR are few, and you can add more primers to give more efficient diversity among the studied populations.

Answer: We partially agree because inclusion of more loci would increase the genomic representation, when selected loci are polymorphic. However to this particular case the evaluation was done with 18 SSR primers from which we obtained 9 polymorphic loci. Furthermore using those primers allow the comparisons between our results and already published studies of closely related species. We also include information that the ISSR evaluation was done with 12 primers of the UBC set 9, from which only four were selected. Adding up the polymorphic loci that was used in this study range from 35-155 loci in the different populations and a total of 155 loci, which guarantee a good representation of the genome.

3-. Request: Results: we need to make a relationship between ISSR and SSR data to give a cluster tree for the 6 populations.

Answer: Done. Figure 3b correspond to a cluster tree for the 6 populations derived from ISSR and SSR data.

4-. Request: Can you send in the revised version Excel sheet for the data from ISSR and SSR used for structure software?

Answer: Attached and included in the manuscript as Supplemented File SF1 (Line 153)

5-. Request: Some corrections in the revised paper in the attachment.

Answer: All corrections have been done.

We thank general and punctual observations that both reviewers made, which improve this manuscript.

Reviewer 2 Report

A review of the manuscript

Genetic variability of Tabebuia rosea (Bignoniaceae) from plantations and remnant populations in the Mayan forest

by

Hugo Ruiz-González, Maria Raggio, Natalia Y. Labrín-Sotomayor, Miriam M. Ferrer, Yuri J. Peña-Ramírez

The paper is devoted to an assessment of genetic variability and genetic differentiation among local populations of a tropical tree Tabebuia rosea (Bignoniaceae) in W Yukatan. The authors used SSR and ISSR markers to study six local populations, three of which represent stands in remnants of a natural forest, and three other represent trees from commercial plantations. In the Introduction the authors mention, that Tabebuia rosea is widely used in Mexico both as a timber tree and as an ornamental. So, the first question arising is whether Tabebuia rosea trees occur in between the populations studied in that area? In other words, are the six studied populations spatially isolated or interconnected by trees in the unsampled area?

The authors analysed their ISSR data using two different Bayesian approaches realized in programs STRUCTURE and BAPS and got different results. They should better describe the models used in these analyses and provide not only log-likelihood but also delta K values and corresponding diagrams obtained in Distruct or Structure Harvester programs. What is the fifth cluster revealed by STRUCTURE and depicted in Fig. 2 with yellow color? It does not correspond to any population and occurs only among admixed individuals. The authors does not interpret it. Another unclear issue is why they did not use their SSR data for the Bayesian analyses? It would be interesting to compare the results from both data sets. I think that the authors’ conclusions about genetic differentiation and bottlenecks are weakly supported by their data given low values of Fst’s and inbreeding coefficients. The Discussion part should be rewritten, because the authors actually discuss general issues but not their own results.

The paper needs language improvement throughout the text. At the present state it is not easy to read.

The language of the manuscript is difficult to read.

Author Response

Reviewer 2:

1-. Request: …the first question arising is whether Tabebuia rosea trees occur in between the populations studied in that area? In other words, are the six studied populations spatially isolated or interconnected by trees in the unsampled area?

Answer: The third paragraph in introduction has been modified including explicitly the information regarding ubiquitous presence of the species in the Yucatan Peninsula (Lines . 56 to58)

2-. Request: The paper needs language improvement throughout the text. At the present state it is not easy to read.

Answer: A professional editing service has been hired to review the English style.

3-. Request: The authors analysed their ISSR data using two different Bayesian approaches realized in programs STRUCTURE and BAPS and got different results. They should better describe the models used in these analyses and provide not only log-likelihood but also delta K values and corresponding diagrams obtained in Distruct or Structure Harvester programs.

Answer: Done. K values and Structure Harvester results are included as Supplementary file SF2 (Line 174). Differences are discussed now in lines 330 to 335

4-. Request: What is the fifth cluster revealed by STRUCTURE and depicted in Fig. 2 with yellow color? It does not correspond to any population and occurs only among admixed individuals. The authors does not interpret it.

Answer: Differences in clustering are mentioned in lines 240 to 269. The clustering in dendrogram in figure 3b now include colors from five clusters (color were changed in figures 2 and 3 for clarity). Results are discussed in lines 443 to454.

5-. Request: Another unclear issue is why they did not use their SSR data for the Bayesian analyses? It would be interesting to compare the results from both data sets.

Answer: We stated in material and method section why we do not perform the genetic differentiation analyses with the SSRs data in lines 175 to 178 “The genetic structure posterior probabilities and the genetic differentiation were not analyzed with the SSRs markers because the high prevalence of null alleles prevents the correct identification of genotypes, that are required to obtain the similarity-differentiation statistics”.

6-. Request: I think that the authors’ conclusions about genetic differentiation and bottlenecks are weakly supported by their data given low values of FST’s and inbreeding coefficient. The Discussion part should be rewritten, because the authors actually discuss general issues but not their own results.

Answer: We thank the reviewer to point out that the FST values were low and therefore geneflow should be high enough to maintain connectivity in this landscape. We rewrite the discussion including all observations, and explaining our results acknowledging that management of native tree plantations may act both as reservoirs of wild populations in the more frequent fragmented landscapes of Tropical ecosystems.

We thank general and punctual observations that both reviewers made, which improve this manuscript.
